# A Series of Avoided Crossings of Resonances in the System of Several Different Dielectric Resonators Results in Giant $Q$-Factors

Konstantin Pichugin , Almas Sadreev * and Evgeny Bulgakov

Kirensky Institute of Physics, Federal Research Center KSC SB RAS, 660036 Krasnoyarsk, Russia
* Correspondence: almas@tnp.krasn.ru

**Abstract:** On an example of a system of three/four/five/six different coupled coaxial silicon disks, we realize a series of avoided crossings of resonances (ACRs) with respect to the different morphologies for the different scales of each disk. Each next step of ACR accompanied by the optimization processes of all previous ACRs contributes almost one order of magnitude to the $Q$-factor. As a result, we achieve unprecedented values for the $Q$-factors: $6.6 \cdot 10^4$ for three, $4.8 \cdot 10^6$ for four, $8.5 \cdot 10^7$ for five and several billions for six free standing silicon disks. Comparisons to such prominent methods as whispering gallery modes or quasi bound states in the continuum to boost the $Q$-factor demonstrate the tremendous advantage of the present approach not only in terms of $Q$-factor values but also in terms of mode volumes. Multipole analysis of the final hybridized resonant mode explains such extremely large $Q$-factor values. The analysis shows a strong redistribution of radiation owing to the almost-exact destructive interference of the dominating complex multipole radiation amplitudes.

**Keywords:** resonant modes; avoided crossing; high $Q$-factor





## 1. Introduction

Since the famous paper by Gustav Mie [1], the engineering of dielectric resonators in optics and photonics has been a long-standing area for the application of various ideas and approaches intended to enhance the quality factor $Q$ due to its paramount importance in both applied and fundamental research. However, there is a fundamental upper limit to the $Q$-factor because of the leakage of radiation power from an isolated dielectric resonator into the radiation continuum [2,3]. There are many ways to enormously boost the $Q$-factor. For example, one can use Fabry–Pérot resonances or hide a resonator in a photonic crystal (PhC) [4–7]. Whispering gallery modes (WGMs) in cavities with convex smooth boundaries, such as cylindrical, spherical or elliptical cavities, also show giant magnitudes of $Q$-factor [8–12].

A cardinally different method originates from bound states in the continuum (BICs) which provide a unique opportunity to confine and manipulate electromagnetic waves within the radiation continuum (see reviews [13–17]).The BIC phenomenon is based on the fact that electromagnetic energy can only leak in selected directions, given by diffraction orders, if the dielectric resonators are arranged in a periodic array [18–20]. Although the number of resonators $N$ in an array cannot actually be infinite, the $Q$-factor grows rapidly with $N$: quadratically for symmetry protected (SP) quasi-BICs [21–23] or cubically for accidental quasi-BICs [22,24,25]. However, this method of engineering quasi-BICs with high $Q$-factor results in dielectric structures (DS) away from compactness. For example, to achieve a $Q$-factor of the order $10^5$, we need at least several tens of silicon disks [23,26,27] or silicon cuboids [28]. The best results for the $Q$-factor were reported by Taghizadeh and Chung [21], with $Q \sim 10^5$ for 10 long identical silicon rods. In general, all of the above methods of achieving an extremely high $Q$-factor require an extended DS in which the mode volume also grows [7,29].

In the this paper, we present a radical method of boosting the $Q$-factor in a system of only a few coupled resonators, each of different scales. Variation of the scales triggers a cascade of avoided crossings of resonances (ACRs), which allows for the drastic increase in the Q-factor. As a result, we have achieved giant magnitudes of the $Q$-factor, significantly exceeding the $Q$-factor of quasi-BICs and maintaining nearly the same mode volume (see Table 1). ACR [30,31] is a general and fundamental phenomenon that describes the behavior of the eigenfrequencies of an open resonator, which are complex due to their coupling with the radiation continuum. Whether the resonant frequencies exhibit either crossing or anticrossing depends on the mechanism of interaction [31–33]. In any case, two resonances near ACR interfere in constructive and destructive ways. The latter means of interference increases the $Q$-factor. As has been demonstrated for different choices of dielectric resonators [34–42], the $Q$-factor can be strongly enhanced in the vicinity of ACR. Example of such a supercavity mode due to the hybridization of resonances is highlighted by the yellow open circle in Figure 1a in the case of a disk-shaped resonator. In a single silicon disk with permittivity $\epsilon = 12$, the $Q$-factor reaches $Q \approx 1.5 \cdot 10^2$ for $h_1/r = 1.4157$, as Figure 1b shows. Along with this, the ACR approach to boosting the $Q$-factor has been developed for systems of photonic molecules owing to coupling between resonators [12,35,43–50]. In particular, strong enhancement of the $Q$-factor has been achieved for two identical coaxial disks [47,51].

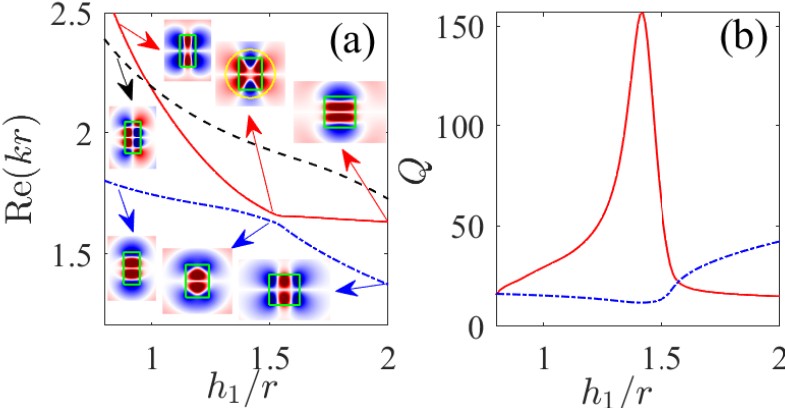

**Figure 1.** (**a**) ACR of the two resonant even modes of a single silicon disk with $\epsilon = 12$ are shown by the red solid line and the blue dashed/dotted line for variation of the aspect ratio [36]. The insets show hybridization of modes (the tangential component of the electric field $E_\phi$ of the TE modes). Disks are outlined by green. The black dashed line shows the evolution of the odd resonant mode, which is decoupled from former modes. (**b**) Respective considerable enhancement of the $Q$-factor due to ACR.

**Table 1.** Mode profiles and parameters of optimized systems of several coaxial disks.

| | Mode Profile Re $(E_\phi)$ | Scales | Re (kr) | $Q$ | $V_m \left(\frac{n}{\lambda}\right)^3$ |
|---|---|---|---|---|---|
| 1 | | $h_1/r = 1.4157$ | 1.72 | $1.5 \cdot 10^2$ | 1.4 |
| 2 | | $h_1/r = 1.257, r_1/r = 1$ $h_2/r = 1.362, L_{12}/r = 0.873$ | 1.76 | $9.8 \cdot 10^3$ | 1.6 |
| 3 | | $h_1/r = 1.292, r_1/r = 1.243$ $h_2/r = 1.375, L_{12}/r = 1.78$ | 1.75 | $5.8 \cdot 10^4$ | 1.6 |

**Table 1.** *Cont.*

| | Mode profile Re ($E_\phi$) | Scales | Re (kr) | Q | $V_m \left(\frac{n}{\lambda}\right)^3$ |
|---|---|---|---|---|---|
| 4 |  | $h_1/r = 0.9972, r_1/r = 1.0363$ <br> $h_2/r = 1.3709, L_{12}/r = 0.8497$ | 1.76 | $6.6 \cdot 10^4$ | 1.4 |
| 5 |  | $h_1/r = 1.0237, r_1/r = 1.0398$ <br> $h_2/r = 1.3025, r_2/r = 1.2319$ <br> $h_3/r = 1.3629, L_{12}/r = 1.5468$ <br> $L_{23}/r = 1.3879$ | 1.77 | $8.5 \cdot 10^7$ | 1.7 |
| 6 |  | $h_1/r = 1.038, L_{12}/r = 0.734$ | 2.19 | $5.7 \cdot 10^3$ | 1.9 |
| 7 |  | $h_1/r = 1.0173, r_1/r = 1$ <br> $h_2/r = 1.039, L_{12}/r = 2.2731$ <br> $L_{23}/r = 0.6585$ | 2.19 | $9.8 \cdot 10^5$ | 2 |
| 8 |  | $h_1/r = 0.7988, r_1/r = 0.8368$ <br> $h_2/r = 1.0503, L_{12}/r = 1.1424$ <br> $L_{23}/r = 0.4922$ | 2.2 | $4.8 \cdot 10^6$ | 2.1 |
| 9 |  | $h_1/r = 1.2236, r_1/r = 1.4054$ <br> $h_2/r = 0.8979, r_2/r = 0.8138$ <br> $h_3/r = 1.0573, L_{12}/r = 0.7712$ <br> $L_{23}/r = 1.0701, L_{34}/r = 0.42213$ | 2.2 | $4.7 \cdot 10^9$ | 2.1 |
| 10 |  | $h_1/r = 0.2588, m = 10$ | 4.76 | $6 \cdot 10^6$ | 4.9 |
| 11 |  | $h_1 = h_2 = h_3 = 1.038r$ <br> $r_1 = r_2 = r$ <br> $L_{12} = L_{23} = L_{34} = 0.734r$ | 2.19 | $2.4 \cdot 10^3$ | 4.4 |

## 2. The Problem Statement

Recently, we have developed a way to enhance the *Q*-factor by extending the number of resonators in photonic molecules spaced at different distances on the example of three and four coaxial silicon disks [52]. As a result, we achieved $Q \approx 10^6$ for four disks of identical radii. In the present paper, we put forward a novel strategy of cascading ACRs in a system of *N* disks in order to achieve unprecedented magnitudes of *Q*-factor. We consider disks to be freestanding and coaxial, made of silicon with permittivity $\epsilon = 12$ for wavelength $\lambda \approx 1.55$ µm, at which material losses are negligible [53]. The eigenmodes of the system are classified according to irreducible one-dimensional representations of rotations around the symmetry axis specified by the azimuthal index *m*. We will focus on the case $m = 0$ since the solutions of Maxwell equations are additionally split by polarization, which also simplifies the problem. A system of *N* coaxial disks offers $3N - 1$ scales to vary, in general, the following: *N* radii $r_j$, *N* heights $h_j$, $j = 1, 2, \ldots, N$, and $N - 1$ distances $L_{12}, L_{23}, \ldots, L_{N-1N}$. Considering that one of the scales should be chosen for dimensionless ratios, we obtain a total of $3N - 2$ parameters. Optimization over this number of parameters, even for a small number of resonatorsm is an extremely time-consuming computational problem. It is reasonable to choose systems that are symmetric with respect to the inversion of the axis of rotational symmetry, which radiates less compared to non-symmetric designs. Guided by this assumption, $N_p = \text{fix}((3N - 1)/2)$ scale parameters are left to vary, where

the term 'fix' means a rounding to the nearest integers towards zero. Particular cases of the systems with $N = 3, 4, 5, 6$ are shown in Figure 2.

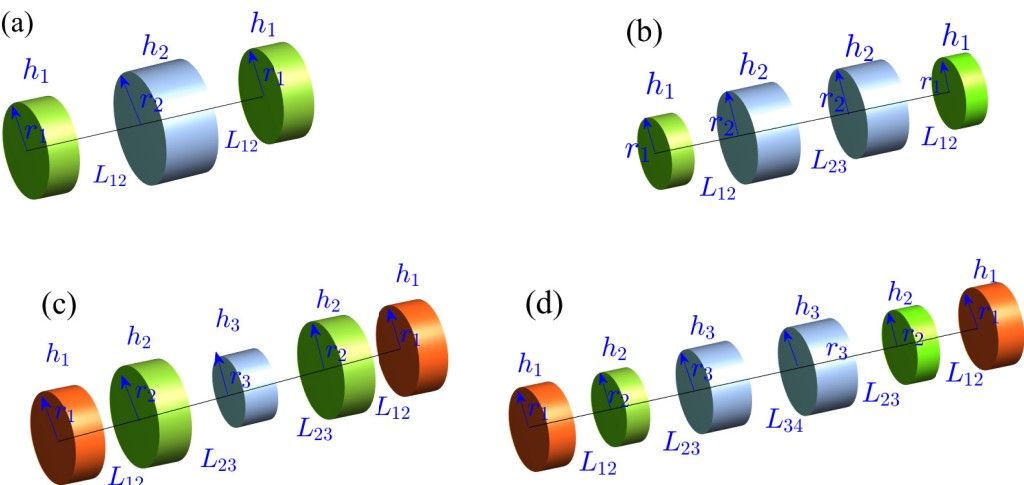

**Figure 2.** (**a**) Disk inside dimer. (**b**) Dimer inside dimer. (**c**) System (**a**) inside dimer. (**d**) System (**b**) inside dimer. All freestanding disks with $\epsilon = 12$ are coaxial but have different radii and heights to form a symmetric structure. The radius of the middle gray disk(s) is used as scale $r$ in the systems. We assume azimuthal number $m = 0$ unless otherwise noted.

Our central approach is based on dividing the system into two subsystems: an internal subsystem of $N - 2$ disks and an external dimer represented by the first and the last disks. We assume that the first internal subsystem has already been optimized to find the hybridized resonant mode $\psi_{N-2}$ with maximal $Q$-factor. This mode could be even or odd with respect to the axis inversion. The outer dimer provides the resonant modes $\psi_2$ of the same symmetry. For variation of scales of the dimer, we have multiple ACRs of its resonances with the optimized resonance of the internal subsystem. As a result, we obtain a hybridized resonant mode $\psi_N$ with enhanced $Q$-factor of the total system. However, it must not be supposed that the solution to the problem is finished. The interaction of two subsystems slightly perturbs the optimized mode $\psi_{N-2}$, which obliges some fine-tuning of the internal subsystem. Therefore, we must continue the process of successive optimizations.

Technically, the strategy appears as follows. To enhance the $Q$-factor, we perform an optimization procedure in parametric space for initial sets of parameters. Each initial set leads to a local maximum of the $Q$-factor. It is reasonable to fix first scale parameters for the internal subsystem of $N - 2$ disks tuned to have maximal $Q$-factors, while the three remaining scales $L_{12}, r_1, h_1$ of the outer dimer evolve in a three-dimensional parametric space. In view of time consuming calculations, we apply the Nelder–Mead simplex optimization method for the $Q$-factor in total $N_p$-dimensional parametric space in conjunction with COMSOL Multiphysics for the finding of complex eigenfrequencies. As a result, we achieve giant magnitudes of the $Q$-factor $6.6 \cdot 10^4$ for three, $8.5 \cdot 10^7$ for five silicon disks at frequency $kr \approx 1.75$, $4.8 \cdot 10^6$ for four, and $4.7 \cdot 10^9$ for six silicon disks at frequency $kr \approx 2.2$, maintaining nearly the same mode volume (see Table 1).

## 3. A Cascade of Avoided Crossings of Resonances in the System of Several Coaxial Disks

*Three Disks*

To illustrate the strategy for achieving maximal $Q$-factor outlined in Section 2, we consider at first the system of three disks sketched in Figure 2a with four independent parameters —$r_1, L_{12}, h_1, h_2$—referring to the radius $r$ of the central disk. First, we place the

already optimized disk $h_2/r = 1.4157$ (see mode 1 in Table 1) between two disks with the same aspect ratio and vary the distance $L_{12}$ only. At a large distance $L_{12}/r$ the resonances are almost degenerate and marked by 'x' in Figure 3a. Drawing closer, the disks interact according to the law $e^{ikL_{12}}/L_{12}^2$ [51] because of the radiation of leaky resonant modes via one disk and subsequent scattering by the others. Couplings between these supercavity modes split them into three modes with spiral behavior as shown in Figure 3a, with three $Q$-factor peaks at corresponding distances as plotted in Figure 3b. Insets show the field configurations of disks related to these $Q$-factor peaks. As a result, we obtain a total gain in the $Q$-factor four times greater than in the case of the single disk marked by a red cross in Figure 3b.

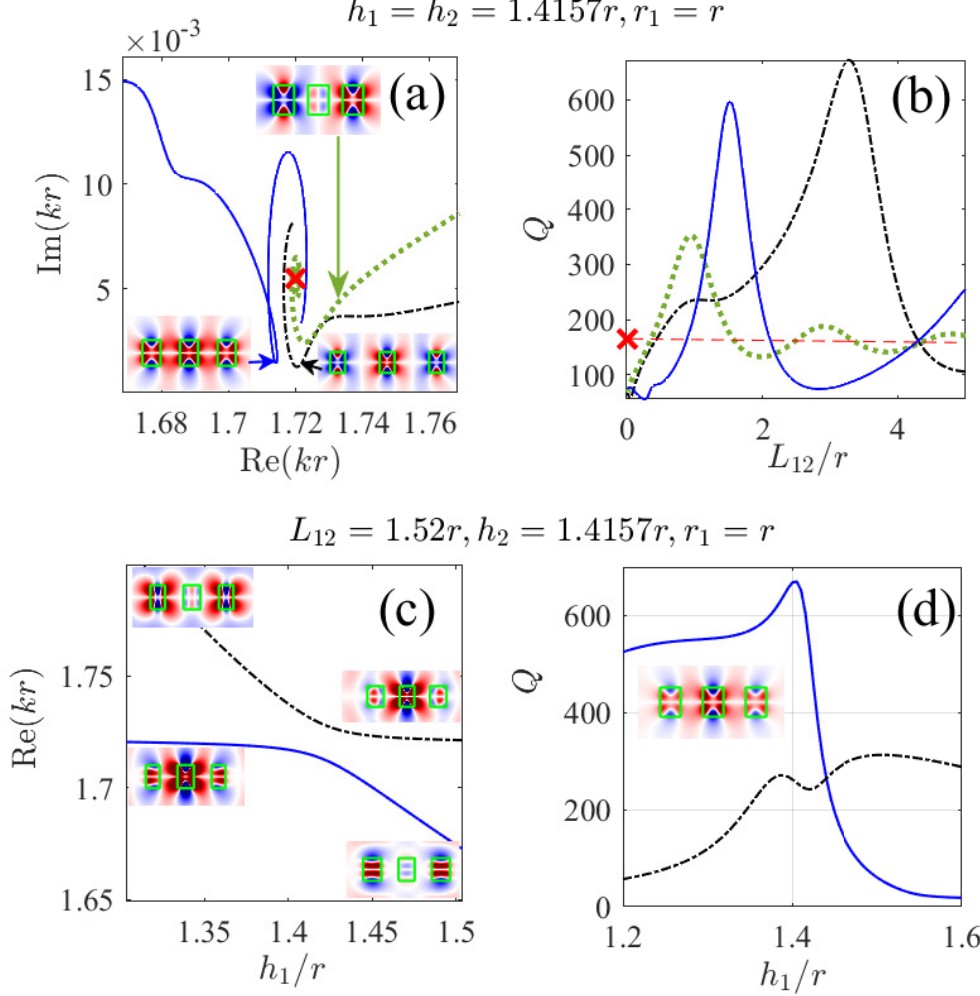

**Figure 3.** (**a**) The first step of optimization over distance $L_{12}$ between three identical disks with fixed parameters $h_1 = h_2 = 1.4153r$, $r_1 = r$. (**b**) Corresponding dependence of the $Q$-factor. The $Q$-factor reaches maxima at $L_{12} = 0.945r$, $Q = 350$ for green line (mode profile $\mathrm{Re}(E_\phi)$ is shown on the middle inset of (**a**)), $L_{12} = 1.525r$, $Q = 600$ for blue line and (mode profile is shown on the left inset of (**a**)), $L_{12} = 3.29r$, $Q = 670$ for black line (mode profile is shown on the right inset of (**a**)). (**c**) The second step of optimization over $h_1/r$ with fixed $L_{12} = 1.52r$, $h_2 = 1.4157r$, $r_1 = r$. The $Q$-factor (**d**) reaches maximum at $h_1 = 1.4031r$, $Q = 670$.

In the next step of the optimization method, we fix $L_{12}/r = 1.52$, at which we obtain the maximum value of $Q = 600$, and vary the height of outer disks $h_1$. This variation gives the ACR of Fabry–Pérot-like mode, which depends strongly on $h_1$, and the Mie-like mode, which depends weakly on $h_1$. This phenomenon, illustrated in Figure 3c, leads to a further enhancement of the $Q$-factor up to $Q = 670$, as shown in Figure 3d.

At the third step, we allow the radius $r_1$ of the outer disks to vary, while all other parameters are fixed to match the maximum $Q$-factor in the second step. In contrast to the previous case, the Mie-like resonant mode of the outer disks is strongly dependent on $r_1$, while the Fabry–Pérot-like mode is weakly dependent on $r_1$. A selected event of ACRs of these modes is shown in Figure 4a, which again raises the $Q$-factor up to 800.

Variation of the height of the inner disk $h_2$, while all other parameters are fixed for $Q = 800$, closes the first-round optimization procedure in full four-dimensional parametric space. The fourth step, as shown in Figure 4c, boosts the $Q$-factor twice compared to the previous step, as seen in Figure 4d. Repeating these rounds, at the end of the optimization procedure, we obtain $Q = 6.6 \cdot 10^4$ (see the mode 4 in Table 1).

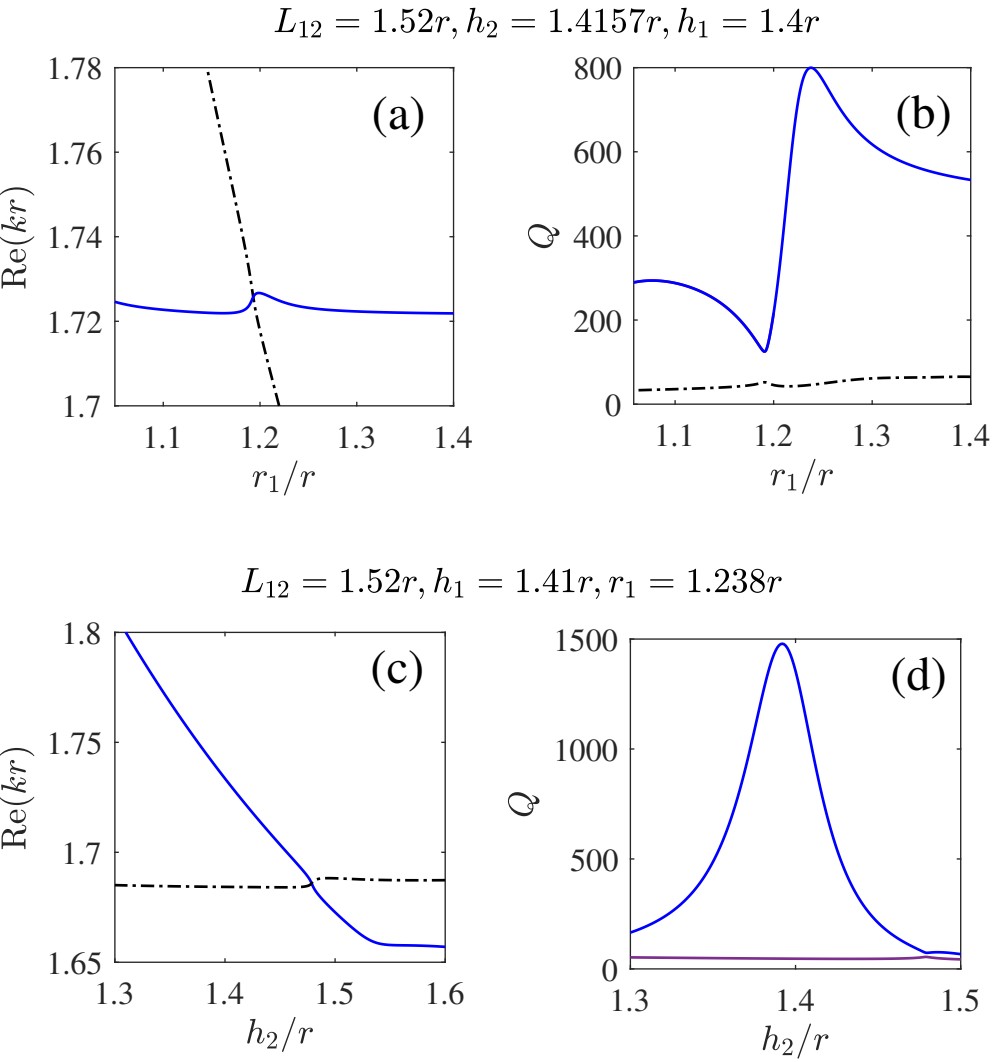

**Figure 4.** Consequent ACRs over variation of different scales in the system of three disks. ACR (**a**) and $Q$-factor (**b**) for variation of the radius $r_1$ of outside disks. ACR (**c**) and $Q$-factor (**d**) for variation of the thickness $h_1$ of outside disks.

Another way to shed light on the enormous enhancement of the $Q$-factor is to see the evolution of resonances along the trajectory obtained by the traditional gradient descent method. This method gives us point $\mathbf{X}_\infty$ in a parametric space with local maximum of the $Q$-factor, which is the limit point of iterations

$$\mathbf{X}_{n+1} = \mathbf{X}_n + \eta \nabla Q(\mathbf{X}_n), n \to \infty, \tag{1}$$

with the appropriately chosen step $\eta$.

The total result of the method can be represented as a trajectory in four-dimensional parametric space, whose length is determined as a curvilinear integral

$$S = \int_{\mathbf{X}_0}^{\mathbf{X}_\infty} \nabla F d\mathbf{s}, \tag{2}$$

where

$$F = \frac{1}{r}\sqrt{h_1^2 + h_2^2 + L_{12}^2 + r_1^2}, \tag{3}$$

and $\mathbf{X}_0$ is an initial point of evolution. The evolution of the three relevant complex eigenfrequencies of the three-disk system is presented in Figure 5 as a function of length $S$. In Figure 5, we can see typical ACR-like behavior: the real parts of the green and blue lines cross (Figure 5b), while the imaginary parts repel each other (Figure 5c). The interaction of at least three eigenfrequencies results in enormous enhancement of the $Q$-factor on the red line.

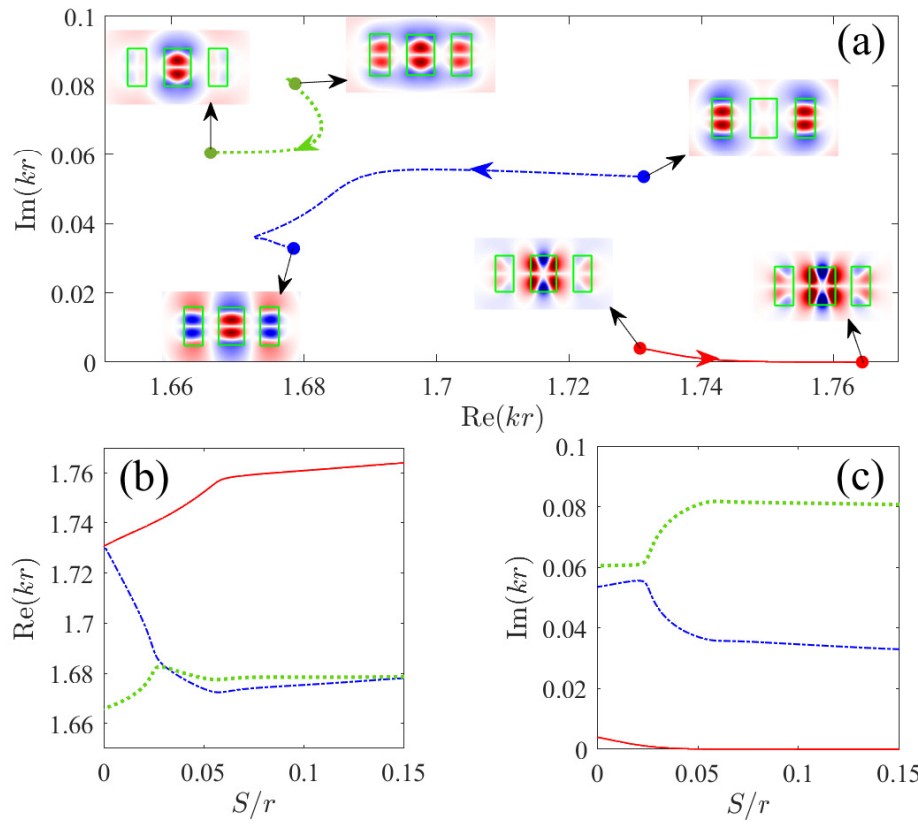

**Figure 5.** (**a**) Evolution (1) of complex eigenfrequencies in the four-dimensional parametric space of all scales in a system of three coaxial disks (**a**). Respectively the real (**b**) and imaginary (**c**) parts of these eigenfrequencies vs the integral $S/r$ (2). $\mathbf{X}_0$ is given by $r_1 = r, h_1 = r, L_{12} = 0.945r, h = 1.4157r$. The final point $\mathbf{X}_\infty$ corresponds to the mode 4 in Table 1.

There are a few points in the full four-dimensional parametric space to which the optimization method converges. In this section, we present only the most outstanding results which exceed the values of the first iteration shown in Figure 4 by several orders of magnitude.

Among the local maxima of the $Q$-factor, there are eigenfrequencies with unprecedented $Q = 5.8 \cdot 10^4$ and $Q = 6.6 \cdot 10^4$, as shown in Figure 6. In both cases, we have very similar supercavity modes in the middle disk, while the structure of the EM field in the outer dimer is different. It is worth noting that the optimized mode for five disks (mode 5 in Table 1) is something of a combination of the above modes; the EM field in disk 1 looks like the EM field in disk 1 of mode 4, while the EM field in disk 2 looks like the EM field in disk

1 of mode 3 in Table 1. Thus, we can conclude that the outer dimer plays an important role in the resonant shielding of the supercavity mode radiation. Because of the even symmetry of the supercavity mode, there are no ACRs of this mode with resonant odd dimer modes.

Next, let us consider the four coaxial disks sketched in Figure 2b. By virtue of inversion symmetry, we represent the system in the form of two dimers: internal and external. In total, we have five scale parameters for ACRs: two heights $h_1$ and $h_2$; the two lengths of the dimers expressed via the two distances $L_{12}$ and $L_{23}$; and, finally, the radius $r_1$ of the outer dimer. All parameters are considered in respect to the radius of the internal dimer $r$.

We omit the iteration steps for all five parameters of the system of two dimers. In addition, the reader can find some scenarios for ACRs for variations of four scales—two heights and two distances for identical radii $r_1 = r$—in our previous publication [52]. This allows us to boost the $Q$-factor up to one million (see mode 7 in Table 1). In the present paper, we perform the final step by optimizing all five parameters.

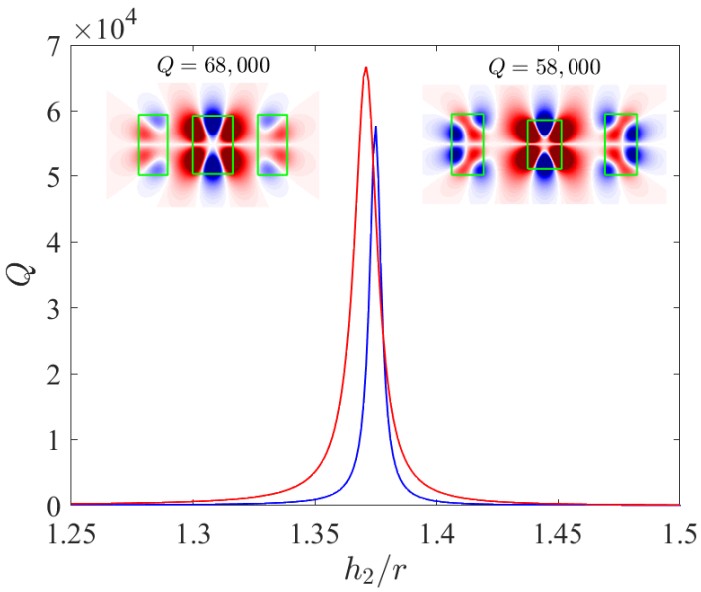

**Figure 6.** Dependence of the $Q$-factor on $h_2/r$ for optimized modes 3 (blue) and 4 (red) in Table 1.

Similar to the case of three disks, shown in Figure 3, we have a Mie-like mode of the outer dimer, labelled as 1 in Figure 7a, which strongly depends on the radius $r_1$ of the external dimer. The other two Fabry–Pérot-like modes, labeled as 2 and 3, are mostly localized in the internal dimer and have weak dependence on $r_1$. As a result, we observe a cascade of ACRs around $r_1/r = 0.8$, highlighted by yellow open circles, which, however, do not lead to magnificent enhancement of the $Q$-factor. In contrast to these conventional ACRs, we observe a slightly noticeable ACR around $r_1/r = 0.85$ with the Fabry–Pérot-like modes of the outer dimer. This results in an enormous boosting of the $Q$-factor up to almost five million, and it can be explained by the cumulative effect of the interaction of the nearby resonances.

In Table, 1 we collected the final configurations of systems of five and six freestanding coaxial silicon disks after the optimization procedure in parametric spaces of dimensions 7 and 8, respectively. It can be seen that the outer dimer almost completely shields radiation from the inner subsystem due to the ACR of the Fabry–Pérot-like resonant mode of the external dimer with the resonant mode of internal subsystem, which has already been optimized for the maximum $Q$-factor. These cases show more impressive $Q$-factor results, i.e., about $5.8 \cdot 10^7$ and one billion (see Table 1). Similar results can be achieved by employing WGM modes in resonators or periodical quasi-BICs, albeit at the cost of increasing the radius or number of identical resonators. This, in turn, increases the mode volume.

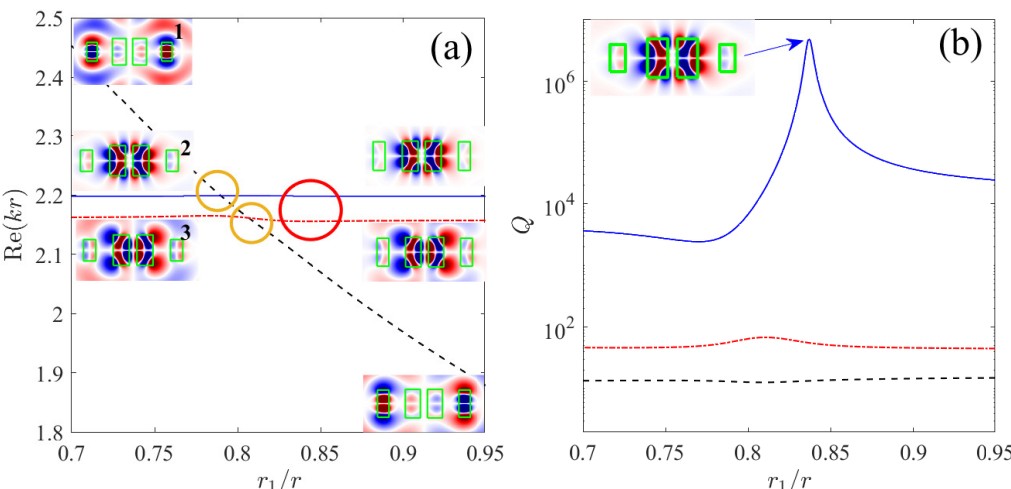

**Figure 7.** (**a**) The final step of ACR for variation of radius $r_1$ of the external dimer relative to the radius of the internal dimer $r$ with strong enhancement of the $Q$-factor (**b**).

## 4. Multipole Radiation for Avoided Crossing of Resonances

There is a useful tool by which to understand the nature of the extremely high quality factor via multipole decompositions [54]. This tool sheds light on the origin of the high $Q$-factor in the isolated disk [38,55] and the origin of bound states in the continuum [52,56]. In the far field region, the EM field can be expanded as

$$\mathbf{E}(\mathbf{x}) = \sum_{l=1}^{\infty} \sum_{m=-l}^{l} [a_{lm}\mathbf{M}_{lm}(\mathbf{x}) + b_{lm}\mathbf{N}_{lm}(\mathbf{x})], \tag{4}$$

where $\overrightarrow{M}_l^m$ and $\overrightarrow{N}_l^m = \frac{1}{k}\nabla \times \overrightarrow{M}_l^m$ are the vector spherical harmonics [57,58]. Then, the relative radiated power of each electric and magnetic multipole of order $l$ is determined by the squares of the decomposition amplitudes: [54]

$$P_{l0} = P_{l0}^{TE} + P_{l0}^{TM} = P_0^{-1}[|a_{l0}|^2 + |b_{l0}|^2], \tag{5}$$

where $P_0 = \sum_{l=1}^{\infty}[|a_{l0}|^2 + |b_{l0}|^2]$ is the total power radiating through a sphere with a large radius. For the present case of coaxial disks with inversion symmetry and azimuthal number $m = 0$, the decomposition (4) is substantially reduced to have an even $l$ for the symmetric solutions shown in Figure 6, and an odd $l$ for the antisymmetric solutions shown in Figure 7 [59].

The extreme $Q$-factor is associated with a strong redistribution of multipole radiation towards high-order multipoles because of the almost-exact total destructive interference of low-order multipole amplitudes. Using the formalism described in Ref. [60] (Equation (1.69)), we separate contributions from subsystems assembling the DS in the far field region. For the case of three disks, we distinguish multipole radiation from the inner disk and outer dimer, whose complex amplitudes $a_{l0}$ in the series Equation (4) are presented in Figure 8.

On subplots (a) and (b), the markers 'o' and 'x' correspond to amplitudes $|a_{l0}|$ of the multipole radiation from the subsystems of the inner disk and outer dimer, respectively, while the red closed circles show the multipole coefficients of the total DS, normalized by $P_0 = \sum_l |a_{l0}|^2 = 1$. Subplots (c) and (d) show the phase difference between the complex amplitudes of the multipole radiation of the subsystems: the inner disk and outer dimer. The left panels of Figure 8 show the case of maximum $Q$-factor $9.8 \cdot 10^3$ achieved by ACR when optimizing three parameters, $h_1$, $h_2$ and $L_{12}$, with $r_1 = r$. One can see strong multipole radiation for $l = 3$ from both the inner disk and outer disks. However, these complex amplitudes $a_{30}$ from both parts, sketched by arrows in the complex plane, have almost the same moduli $|a_{30}|$ and a phase difference close to $180°$, which results in the nearly full

destructive interference of multipole radiation at $l = 3$. The total multipole radiation of the DS for $l = 3$ vanishes, as shown by the red closed circle in Figure 8a. Note that there is still small multipole radiation at $l = 7$ from the outer dimer, while the radiation from the inner disk is mostly suppressed.

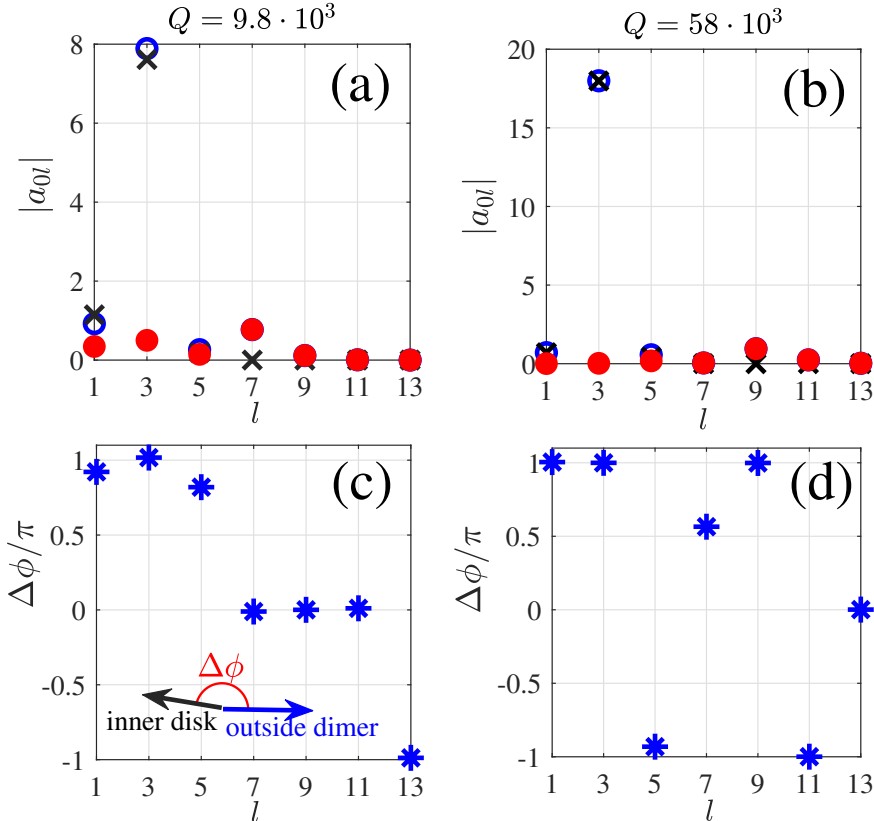

**Figure 8.** The multipole radiation amplitudes $a_{l0}, l = 1, 3, 5, \ldots$ in Equation (4) from the system of three disks of identical radii (mode 2 in Table 1, $Q = 9.8 \cdot 10^3$) (**a,c**) and of different radii (**b,d**) (mode 3 in Table 1, $Q = 5.8 \cdot 10^4$). The amplitudes of the inner/outer dimer disk marked by crosses/open circles and the amplitudes of the total system of three disks are marked by red closed circles. Black and blue arrows show complex radiation amplitudes with phase difference $\Delta\phi$.

Let us now consider the right panels of Figure 8, which show multipole radiation with $Q = 5.8 \cdot 10^4$ achieved by optimizing all possible parameters: $h_1$, $r_1$, $h_2$ and $L_{12}$. We can see from Figure 8b that the additional optimization over $r_1$ shifts the channel of maximum radiation from $l = 7$ to $l = 9$ and increases the $Q$-factor by six times. One can speculate that the introduction of an additional parameter could suppress more contributions into multipole radiation. For example, two dimers, as shown in Figure 2b, provide more geometrical parameters than the previous case presented in Figure 2a. The amplitudes of the multipole decomposition are shown in Figure 9. We can see the remarkable effect of the destructive interference of complex multipole amplitudes $a_{l0}$. The left panels demonstrate the effect till $l = 4$, while the right panels do so till $l = 14$.

In subplots (c) and (d), the relative phases between amplitudes of both dimers are shown. One can observe in Figure 9 the almost-full destructive interference of the multipolar amplitudes at the dominant channels $l = 2, 4, 6$ from both dimers when moduli of the coefficients are equal, while phases differ by $\pi$. The destructive interference of several amplitudes $|a_{0l}|$ simultaneously was achieved owing only to the multiscale optimization procedure.

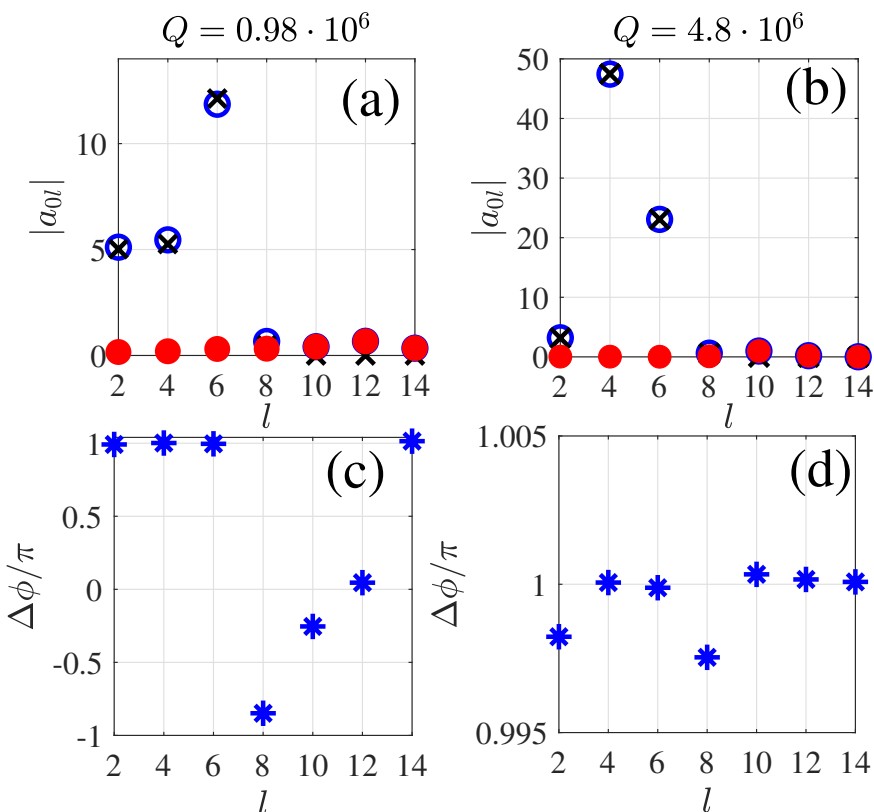

**Figure 9.** The multipole radiation amplitudes $a_{l0}$, $l = 2, 4, 6, \ldots$ in Equation (5) for the system of four disks with identical radii (mode 7 in Table 1, $Q = 9.8 \cdot 10^5$) (**a**,**c**). (**b**,**d**) amplitudes $a_{l0}$ for the system of four disks with different radii (mode 8 in Table 1, $Q = 4.8 \cdot 10^6$). The amplitudes of the inner/outer dimer of the disks are marked by crosses/open circles, and amplitudes of the total system are marked by red closed circles.

## 5. Mode Volumes of Resonances with Extremal *Q*-Factor

Optical cavities are able to trap light at discrete resonant frequencies in a tiny volume in which the interaction of light with matter can be dramatically enhanced via the temporal and spatial confinement of light. It is important not only to enhance the *Q*-factor; the miniaturization of resonators with a high $Q/V_m$ ratio is in demand in order to improve the light–matter interaction and reduce the layout for compact integrated optical circuits.

The *Q*-factor and effective mode volume $V_m$ —two figures of merit in optical cavities—are of great importance in the enhancement of the light–matter interaction. The mode volume of a dielectric resonator is given by the ratio of the total electric energy to the maximum electric energy density [61]

$$V_m = \frac{\int \epsilon(\mathbf{x}) |\mathbf{E}(\mathbf{x})|^2 dV}{\max[\epsilon(\mathbf{x}) |\mathbf{E}(\mathbf{x})|^2]}. \tag{6}$$

A summary of the *Q*-factors and mode volumes is collected by Vahala [4] and range from $Q = 2 \cdot 10^3$, $V_m = 5(\lambda)^3$ (FPR), $Q = 1.2 \cdot 10^4$ and $V_m = 6(\lambda)^3$ (WGM) till $Q = 1.3 \cdot 10^4$, $V_m = 1.2(\lambda)^3$ (PhC cavity). Ultra-low mode volumes in one-dimensional slotted photonic crystal single silicon nanobeam cavities of the order $(0.1 - 0.01)(\lambda/n)^3$ ($n$ is the refractive index of DS) were reported [5–7], albeit at the cost of the compactness of the resonator.

These data in Table 1 are compared with the whispering gallery mode, with azimuthal number $m = 10$, and the eigenfrequency $\mathrm{Re}(kr) = 4.76$ in a single disk of aspect ratio $h_1/r = 0.2588$.

## 6. Conclusions and Outlook

The avoided crossing of resonances lead to the substantial redistribution of their imaginary parts and the hybridization of resonance modes [31]. This method of increasing

the $Q$-factor proved to be successful even in a single resonator in the form of a disk [36] or of a long rod with a rectangular cross-section [42]. ACRs in two identical resonator raises the $Q$-factor significantly more [34,43,51,62]. It would seem that a further increase in the number of identical resonators $N$ is the best way to boost the $Q$-factor since the periodic array of resonators supports quasi-BICs [23] with asymptotic $Q \sim N^2$. However, this method of enhancement of the $Q$-factor bumps into saturation owing to material losses [23] and structural fluctuations [63]. Moreover, quasi-BIC modes concede the compactness of DS and the mode volume. In the present paper, we show that a DS composed of resonators with different scales provides a significantly higher $Q$-factor, preserving nearly the same mode volume as displayed Table 1. In addition, what is remarkable is that these unprecedented $Q$-factors refer to compact DSs, which are fundamentally different from the extended periodic DSs that support quasi-BICs. The compactness of a high-$Q$ DS bestows a great technological advantage with respect to sensing and lasing devices.

To understand the nature of ACRs which lead to extremely high quality factors, one can use multipole decomposition [54]. This tool sheds light on the origin of the high $Q$-factor in the isolated disk [38,55] and the origin of bound states in the continuum [56]. In the present case of several resonators, we also observe that the extreme $Q$-factor is associated with a strong redistribution of radiation and arises from the compensation of dominat multipole coefficients. Moreover, we show that it is related to the almost-perfect destructive interference of the low-order multipole radiations from the internal subsystem inserted into the external dimer.

Thus, the way to boost the quality factor of an array of resonators looks simple. First, we attain the maximum $Q$-factor via ACRs in the internal subsystem of $N - 2$ disks, which results in a hybridized resonant mode. This mode $\psi_{s,a}(N - 2)$ can be symmetric or antisymmetric relative to axis inversion. Then, we symmetrically enclose the internal subsystem into a shell consisting of two identical disks forming an external dimer. Varying the scales of the dimer (radius $r_1$, height $h_1$ and distance $L_{12}$), we trigger ACRs of resonant modes $\psi_{s,a}(N - 2)$ with the resonant modes $\psi_{s,a}(2)$ of the external dimer. To achieve extremal $Q$-factors, one has to allow a slight change of scales in the internal subsystem too, as was shown in Section 3. Because of this, the optimization procedure should be performed over all scales of the total system. As a result, we can achieve almost-perfect shielding of the internal resonant mode by the external dimer and boost the $Q$-factor by several orders of magnitude. Some hybridized resonant modes are collected in Table 1. Our research shows that the multiscale optimization procedure grants substantially higher $Q$-factor results compared to the case of equidistant identical disks, which support quasi-BICs near $\Gamma$-point. The proposed algorithm could be easily adapted to multi-particle systems of different shapes and permittivities. The present system of coaxial disks was chosen because of the separation of TE and TM polarizations in zero azimuthal index sector. It should be noted that the $Q$-factor of optimized systems is very sensitive to scale parameters, aspecially to parameters which affect the resonant mode $\psi_{N-2}$ of the internal subsystem because of the weak localization of the total mode $\psi_N$ at the outside dimer. This brings definite technological problems because of the necessity og accurately setting up the different scales of different resonators.

Fortunately, S. Kim et al. [64] reported control over the disk dimensions with an accuracy of about 5 nm in optical range, i.e., $\sim$0.3% relative to the optical wave length $\lambda = 1.55$ μm.

**Author Contributions:** Conceptualization, A.S. and K.P.; methodology, K.P.; software, K.P. and E.B; validation, all authors; formal analysis, K.P.; investigation, all authors; writing—original draft preparation, A.S.; writing—review and editing, all authors; visualization, A.S. and K.P.; supervision, A.S. All authors have read and agreed to the published version of the manuscript.

**Funding:** This work is supported by the Russian Science Foundation under grant 22-12-00070.

**Data Availability Statement:** Data is contained within the article

**Acknowledgments:** We acknowledge discussions with Yi Xu.

**Conflicts of Interest:** The authors declare no conflict of interest regarding this article.

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
