# Peer review of "A Series of Avoided Crossings of Resonances in the System of Several Different Dielectric Resonators Results in Giant Q-Factors"

_photonics, doi:10.3390/photonics10090973_

Round 1

Reviewer 1 Report

For dielectric resonator, the Q factor is a key parameter for novel physics and advanced application. In contrast to conventional method with resonator arrays to boost quality factor, this manuscript presents a novel way that can get rid of large array numbers. The physical insight of ACR is clearly illustrated and the results are confirmed unambiguously with a couple of models. With that I would like to recommend this paper to be published as present.

Typo in line 259: That tool sheds light on...

Author Response

Referee 1:
For dielectric resonator, the Q factor is a key parameter for novel physics and advanced application. In contrast to conventional method with resonator arrays to boost quality factor, this manuscript presents a novel way that can get rid of large array numbers. The physical insight of ACR is clearly illustrated and the results are confirmed unambiguously with a couple of models. With that I would like to recommend this paper to be published as present.
---------------------------------------------------------
Author response:
We grateful the Referee for his\her high evaluation of our article.

Reviewer 2 Report

The authors present theoretical calcualtions of optical modes in arrays of disks serving as Mie scatterers. By adjusting the geometrical parameters of the scatterers they obtain destructive interference of the electric field from individual scatteres resulting in Fabry-Perot like resonators with high Q-factors. The paper is basically a continuation of the authors previous paper [53], the only difference is that in this paper they improve the optimization mehtod they used in [53] and apply it to an increasingly complex system. The paper is well written and appropriate for publication in Photonics journal. There are some minor issuses:

1)      The authors don't specify how they made the calculations. If they used a comercial modeling software they should include this information

2)      Some of the formating of the Figures and equations is a bit out of bounds of the page limits, in particular Table 1 and equation on line 186 should be rescaled to fit on the page.

 Although the english in the paper is good for the most part, in some sections it is not of the same quality. In particular in the sections „Introduction“ and „Conclusions and outlook“ the quality of english is of lower quality than in the other section and should be improved, especially in the couple of sentences about BIC. 

Author Response

Referee 2:
The authors present theoretical calcualtions of optical modes in arrays of disks serving as Mie scatterers. By adjusting the geometrical parameters of the scatterers they obtain destructive interference of the electric field from individual scatteres resulting in Fabry-Perot like resonators with high Q-factors. The paper is basically a continuation of the authors previous paper [53], the only difference is that in this paper they improve the optimization mehtod they used in [53] and apply it to an increasingly complex system. The paper is well written and appropriate for publication in Photonics journal. There are some minor issuses:

1)      The authors don't specify how they made the calculations. If they used a comercial modeling software they should include this information

2)      Some of the formating of the Figures and equations is a bit out of bounds of the page limits, in particular Table 1 and equation on line 186 should be rescaled to fit on the page.

Comments on the Quality of English Language
 Although the english in the paper is good for the most part, in some sections it is not of the same quality. In particular in the sections „Introduction“ and „Conclusions and outlook“ the quality of english is of lower quality than in the other section and should be improved, especially in the couple of sentences about BIC. 
---------------------------------------------------------
Author response:
We would like to thank the Referee for a thorough review of our manuscript.
All remarks are taken into account:
1. We included reference to COMSOL Multiphysics near the end of the Section 2.
2. Formating issues are eliminated.
3. We tried to improve a quality of English in Sections „Introduction“ and „Conclusions and outlook“.

Reviewer 3 Report

The authors reported ultrahigh-Q resonances in several different dielectric resonators by harnessing avoided crossing of resonances. Through judicious design, the Q-factors for three, four, five, six cylinders can be up to 6.6*10^4, 4.8*10^6, 8.5*10^7 and 10^9, respectively. Besides, the mode volumes are also significantly improved. Such phenomenon are well explained from multipole decomposition analysis. The results in this work are very interesting and novel enough to deserve publications in photonics. I only have several minor concerns

(1)   The authors studied the case of N cylinders (N>2). What about N=2 ?

(2)   Reference 18 is not a Review paper, please remove it from [13-18]. I may suggest authors adding one of the recent review [L. Huang et al, Physics Reports 1008, 1-66 (2023)].  

Author Response

Referee 3:
he authors reported ultrahigh-Q resonances in several different dielectric resonators by harnessing avoided crossing of resonances. Through judicious design, the Q-factors for three, four, five, six cylinders can be up to 6.6*10^4, 4.8*10^6, 8.5*10^7 and 10^9, respectively. Besides, the mode volumes are also significantly improved. Such phenomenon are well explained from multipole decomposition analysis. The results in this work are very interesting and novel enough to deserve publications in photonics. I only have several minor concerns

(1)   The authors studied the case of N cylinders (N>2). What about N=2 ?

(2)   Reference 18 is not a Review paper, please remove it from [13-18]. I may suggest authors adding one of the recent review [L. Huang et al, Physics Reports 1008, 1-66 (2023)].  
---------------------------------------------------------
Author response:
We thank the Referee for the positive evaluation of our paper.
1. We added citation of articles where two identical cylinders are considered at the end of the Section 1.
2. The Reference 18 of former manuscript is removed.